# An Exploratory Study of Large-Scale Brain Networks during Gambling Using SEEG

**DOI:** 10.3390/brainsci14080773

**Published:** 2024-07-31

**Authors:** Christopher Taylor, Macauley Smith Breault, Daniel Dorman, Patrick Greene, Pierre Sacré, Aaron Sampson, Ernst Niebur, Veit Stuphorn, Jorge González-Martínez, Sridevi Sarma

**Affiliations:** 1Department of Biomedical Engineering, Johns Hopkins University, Baltimore, MD 21218, USA; chris.j.w.taylor94@gmail.com (C.T.); daniel.b.dorman@jhu.edu (D.D.); pagreene@jhu.edu (P.G.); ssarma2@jhu.edu (S.S.); 2The Picower Institute for Learning and Memory, Massachusetts Institute of Technology, Cambridge, MA 02139, USA; 3Department of Electrical Engineering and Computer Science, University of Liège, 4000 Liège, Belgium; p.sacre@uliege.be; 4Solomon Snyder Department of Neuroscience, Johns Hopkins University, Baltimore, MD 21218, USA; asamps10@jhu.edu (A.S.); niebur@jhu.edu (E.N.); veit@jhu.edu (V.S.); 5School of Medicine, University of Pittsburgh, Pittsburgh, PA 15213, USA; gonzalezmartinezja@upmc.edu

**Keywords:** decision-making, gambling, intracranial EEG, functional connectivity, default mode network, dorsal attention network, frontoparietal network

## Abstract

Decision-making is a cognitive process involving working memory, executive function, and attention. However, the connectivity of large-scale brain networks during decision-making is not well understood. This is because gaining access to large-scale brain networks in humans is still a novel process. Here, we used SEEG (stereoelectroencephalography) to record neural activity from the default mode network (DMN), dorsal attention network (DAN), and frontoparietal network (FN) in ten humans while they performed a gambling task in the form of the card game, “War”. By observing these networks during a decision-making period, we related the activity of and connectivity between these networks. In particular, we found that gamma band activity was directly related to a participant’s ability to bet logically, deciding what betting amount would result in the highest monetary gain or lowest monetary loss throughout a session of the game. We also found connectivity between the DAN and the relation to a participant’s performance. Specifically, participants with higher connectivity between and within these networks had higher earnings. Our preliminary findings suggest that connectivity and activity between these networks are essential during decision-making.

## 1. Introduction

Cognitive psychology describes decision-making as a cognitive process entailing deliberate and thoughtful choices by gathering, storing, organizing, and evaluating information about options [1]. During economic decision-making in particular, participants with greater cognitive control tend to earn more money [2]. Three large-scale brain networks—the dorsal attention network (DAN), default mode network (DMN), and frontoparietal network (FN)—have been shown to relate to cognitive constructs during decision-making [3,4]. In this preliminary study, we sought to investigate the role of these networks in the context of an economic decision-making task of gambling.

During gambling, one must direct one’s focus toward stimuli that will aid in the gathering of useful information. The DAN has been implicated in selective visual attention by suppressing distractions to focus on the stimuli of interest based on the stimuli’s relevance [5,6]. It anatomically consists of the frontal eye fields and the posterior parietal cortex, which are connected by the superior longitudinal fasciculus [5,6]. The DAN is thus engaged in visual processing and interpretation. For example, an fMRI study found that the DAN has higher activation when relevant visual targets are co-presented with high-impact distractors and are associated with the color or location of objects or if participants are attending to a single target [5]. Further, lesions to the DAN can cause patients to experience difficulties in controlling their spatial orientation, particularly due to an impaired ability to filter out irrelevant distractions based on their current action goals [6]. Specific frequency bands, namely theta and gamma, are also implicated in the functionality of DAN. One study showed that theta band (4–8 Hz) functional connectivity in the DAN predicted the impairment of goal-directed processing in stroke patients with spatial attention deficits using a resting-state EEG [7]. Another intracranial EEG study found sustained power in broadband gamma (50–150 Hz) increased during a search task across the entire DAN [8].

In addition to selective attention, decision-making is affected by working memory and cognitive reasoning as one evaluates the potential outcomes of their possible choices. Cognitive reasoning is the ability to relate one’s previous experiences to completing a task [9]. The DMN is a set of widely distributed brain regions spanning the parietal, temporal, and frontal cortex that have been implicated in cognitive reasoning [10,11]. For example, one fMRI study showed increased BOLD activity in the DMN when participants made decisions based on previous experience (i.e., stored representations of the environment) versus when individuals made decisions based only on the current options [12]. The gamma band is also strongly implicated in the DMN. An intracranial EEG study observed transient suppression in the DMN in the broadband gamma band during a cognitively complex visual search task, which itself was correlated to their performance [13]. In other words, participants reacted more slowly (performed worse) and had less gamma suppression in the DMN for more complex trials.

Finally, gambling requires participants to adapt to the circumstances presented to them. The FN, also known as the central executive network, is linked to flexible cognitive control by harnessing working memory and cognitive reasoning [14]. The FN consists of the inferior frontal junction, lateral prefrontal cortex, anterior cingulate cortex, and posterior parietal cortex [15,16]. The FN has also been identified in relational thinking, specifically numerical thinking, i.e., determining whether one value is greater than another value [17,18]. One scalp EEG study suggested that theta power and connectivity in the FN relate to working memory capacity and attention control [19]. Another study used positron emission tomography (PET) during a rapid visual information processing task where participants had to respond when identifying a sequence of numbers [20]. Researchers found that participants relied on the FN to maintain selective attention and on working memory to determine where a sequence began.

Together, these networks orchestrate their activity through direct connections between them and through intermediate networks. For example, all three networks have been found to modulate their activity individually and collectively based on the dynamics of an internal state as dictated by task performance [21]. Studies using resting-state functional connectivity have identified increased connectivity between the DMN and FN in individuals who are greater risk-seekers, highlighting the importance of their relationship during affective decision-making [22]. Interestingly, a disruption in connectivity between these regions has been identified in people with traumatic brain injury, which is marked by deficient higher-order cognitive functions such as decision-making [23].

While several studies have investigated activity in large-scale brain networks during decision-making, there is still an unclear relationship between activity and connectivity within and between these networks in the context of performance during economic decision-making. Most studies investigating the activity and connectivity of these networks have relied on non-invasive imaging techniques such as fMRI. However, fMRI has poor temporal and spatial resolution [24]. Modern recording techniques, such as stereoelectroencephalography (SEEG), though invasive, are superior in both categories by recording close to the neuronal source with high temporal resolution. Of note is a recent study that used the enhanced spatial and temporal resolution of SEEG to uncover how subjective value during decision-making is represented by gamma activity in the orbitofrontal cortex [25]. Similarly, we sought to utilize SEEG with broader coverage to address network activity during decision-making.

In this study, we used a gambling task to explore the relationship between activity and functional connectivity between the DAN, DMN, and FN with performance during an economic decision-making task in humans using SEEG. Participants played a computer-simulated gambling card game called “War”, in which they decided whether to bet high ($20) or low ($5) based on their belief of whether the value of their known card was greater than the computer’s unknown card in each trial [26]. Simultaneously, we recorded the local field potential activity from the intracranial depth electrodes across a broad range of brain regions that participants were implanted with for clinical purposes. Using earnings as a metric for participant performance, we investigated neural theta and gamma bands activity in the three networks (DAN, DMN, FN) while participants performed the task. Here, we describe the preliminary evidence that relates participant performance to activity in these bands as well as the connectivity within and between these three networks. Specifically, we were interested in investigating the following: (i) describe the relationship between betting behavior and performance across participants, (ii) identify frequency bands in which networks relate to gambling performance, and (iii) describe the relationship between connectivity and gambling performance.

## 2. Materials and Methods

### 2.1. Participants

Intracranial SEEG recordings were taken from ten human participants while they performed an economic decision-making task at Cleveland Clinic. All participants (seven females, three males, average age of 36 years old) were diagnosed with medically refractory epilepsy and were undergoing seizure monitoring using surgically implanted depth electrodes. The Cleveland Clinic Institutional Review Board approved all experimental protocols and procedures as they were conducted as per their guidelines and regulations. Patients were identified by clinicians for eligibility to participate in this study. Research staff, who were not involved in their clinical care, would approach patients with the opportunity to participate in this study. Consent was obtained if the patients expressed interest in participating. Detailed information about patient enrollment has been outlined by Johnson et al. [27]. Participation was completely voluntary, and all participants gave informed consent. The inclusion criteria required all participants to be 18 years or older and able to provide informed consent. No changes were made to the participants’ clinical care beyond them performing the decision-making task. Participants were not taking their epilepsy medications during seizure monitoring for clinical reasons, so the effects of anti-seizure medication on neural activity were minimized. Table 1 contains the demographic and clinical information, such as the diagnosed epileptogenic zone (EZ), for each participant. These data were also used in our previous publications [21,26,27,28,29].

### 2.2. Electrophysiological Recordings

Each participant underwent surgical implantation of 9 to 14 stereotactically placed depth electrodes (PMT Corporation, Chanhassen, MN, USA) for clinical purposes. Each electrode had 10 to 16 electrode contacts; each contact was 0.8 mm in diameter, 2 mm long, and 1.5 mm away from the next contact. The depth electrodes were implanted using a robotic surgical implantation platform (ROSA, Montpellier, France) in orthogonal or oblique configurations to target lateral, intermediate, and deep cortical and subcortical structures [30]. Electrophysiological data were recorded with the Neurofax EEG-1200 system (Nihon Kohden, Irvine, CA, USA) at a sampling rate of either 1 or 2 kHz using the skull as a reference. The behavioral data were synchronized with the electrophysiology data for offline analysis. Neither the behavioral nor electrophysiological data were filtered prior to offline analysis. The anatomical location of each electrode contact for each patient was identified by two clinical experts through post-operative imaging using labels from the Destrieux atlas [31]. The location and number of electrodes used in this study are detailed in Table 1.

### 2.3. Gambling Task

Participants completed a simplified gambling task based on the card game, “War”, where they played for virtual money against the computer. Cards for both the participant and the computer were drawn from a uniform infinite deck consisting only of cards 2, 4, 6, 8, and 10. This task was run using a MATLAB-based (R2012b, MathWorks, Natick, MA, USA) software tool called MonkeyLogic (v2.0) [32] on a Windows-based computer that was taken to the participant’s room in the Epilepsy Monitoring Unit at the Cleveland Clinic. Participants were presented with the visual stimuli of the gambling task on a computer screen (Figure 1A), which they could interact with by using an InMotion2 robotic manipulandum to move a cursor on the screen (Interactive Motion Technologies, Watertown, MA, USA) [33]. The task was broken down into epochs. Each trial was initiated when the participant centered their cursor onto a white icon in the center of the screen within 8 s (“Fixation”). Participants were then shown the face value of their card and the (non-informative) back of the computer’s card for 2 s (“Show Player Card”). The participant was then instructed to bet either $5 (low) or $20 (high) that their card would be higher than the computer card by moving the cursor to their bet; bets were displayed as circles on the computer screen (“Show Bet”). They had 6 s to make their decision. The time when the participant started to move the manipulandum to select their bet was defined as the “Start Movement”. After selecting the bet, the face value of the computer’s card was revealed (“Show Deck”), followed by a screen that displayed the outcome of the trial (“Show Reward”). The following three outcomes were possible: If the participant’s card was higher, then the participant won the amount they bet; if the computer card was higher, then the participant lost the amount they bet; and if the cards were equal, then the trial was declared a draw where no money was received or lost. Participants performed the task for approximately 30 min and completed an average of 163 ± 14 (mean ± 1 standard deviation) trials following an initial brief practice period. Their cumulative earnings were never displayed to the participant during the session. We calculated the cumulative earnings as the summation of bets won and lost throughout the entire session. The average earnings were found by taking the average of the bets won (positive) and lost (negative) across all trials in the session. Participants could be combined into groups of high and low earners, using the population median as the criterion. Reaction time was taken as the time between “Show Bet” and “Start Movement” in seconds. Table 2 contains the performance-related information for each participant. Figure 1B–D shows the cumulative earnings, betting strategy, and average reaction times for two participants (lowest and highest earners). Figure 1E shows the types of neural coverage and electrophysiological recordings during a session, and Figure 1F shows the spectral power in the gamma band for one region for the same two participants.

### 2.4. Large-Scale Brain Networks

The electrode contact locations, manually labeled by clinicians based on the Destrieux atlas, were converted to the Yeo atlas, a common atlas for large-scale brain networks [11], to perform a network-based analysis. This was performed by mapping each voxel on the FreeSurfer template brain called cvs_avg35_inMNI152 [34] from the Destrieux label into a Yeo label, where each region from the Destrieux atlas is the combination of all the voxels with the same label. Therefore, each region with the same Destrieux label consisted of some composition of labels from the Yeo atlas. Regions were considered part of a network if more than 50% of the voxels within the region were contained within the same network. The networks investigated for this study were the Default Mode Network (DMN), Dorsal Attention Network (DAN), and Frontoparietal Network (FN) as shown in Figure 2.

### 2.5. Electrophysiological Preprocessing

The electrophysiological data were preprocessed by subtracting a 10 s moving average on each electrode contact to reduce voltage drift after filtering data using a bandpass filter between 0.5 and 200 Hz. Additionally, 60 Hz electrical noise and its harmonic (120 Hz) were filtered out using a second-order Butterworth notch filter. Electrode contacts with abnormal activity, epileptic activity, or those located in the EZ were removed from further analysis. We calculated the power in six frequency bands—delta (0.5–4 Hz), theta (4–8 Hz), alpha (8–12 Hz), beta (12–30 Hz), low gamma (30–70 Hz), and high gamma (70–150 Hz)—using the MATLAB bandpower function (Signal Processing Toolbox) applied to a moving window of width 500 ms (100 ms for high gamma). The window was shifted by 10 ms for each estimate. A Hamming window was applied to suppress band leakage. Power values at contacts within each region (e.g., anterior or posterior cingulate cortex) were combined by averaging.

### 2.6. Compute Neural Activity within Networks during Decision-Making

The frequency band powers were normalized by applying the natural logarithm and then computing the z-score for each electrode contact using the entire session. Since we were interested in the neural activity during decision-making, we used a window of time between the start of “Show Player Card” to the end of “Start Movement” for our neural analysis. We refer to this time as the decision-making period. The frequency band normalized powers of the electrode contacts within the same region were averaged together, so each region would be represented by one signal per frequency band. Finally, the collection of regions in each of the large-scale brain networks was averaged together, resulting in each network having a signal of normalized powers during the decision-making period for each frequency band. This process was repeated for each participant. Note that participants may not have all regions within a network covered. For example, 9 out of 10 participants had electrodes in the posterior cingulate cortex (PCC) for the DMN. See Table 1 for the complete list of regions within each network per participant.

### 2.7. Compute Connectivity within and between Networks during Decision-Making

The first step in our connectivity analysis was to calculate the connectivity between regions. This was carried out by taking the covariance between the normalized frequency band power (from Section 2.6) between all pairs of regions, using the same time window between the “Show Player Card” and “Start Movement”, i.e., the decision-making period. To compute the connectivity within and between large-scale brain networks, we took the Frobenius norm of the covariance matrix between regions for each pair of networks. We defined this value as the connectivity strength. Statistical comparisons of connectivity strength between low and high performers were calculated for each frequency band and network pair using Bonferroni-corrected Mann–Whitney U tests, and statistical power was assessed from the sample size, U test statistic, and effect size. Effect sizes were calculated as 1 − (2U)/(n1∗n2) (see Appendix A). We also calculated the sign of the connectivity by averaging the sign of the correlation for each frequency band and network pair per participant (see Appendix A).

## 3. Results

### 3.1. Betting Behavior and Performance Varies across Participants

Participants implemented various strategies as they played the game. To highlight this, we examined the betting pattern of participants with the highest (participant 8, green) and lowest (participant 4, orange) average earnings across trials (Figure 3A). Participant 8 made better decisions throughout their session, in the sense that they bet more consistently to maximize the expected reward for each card. As shown in Figure 3B, participant 8 primarily bet low on the 2, 4, and 6 cards and high on 8 and 10 cards. Participant 4, on the other hand, appeared to be changing strategies, i.e., making different decisions on the same card values over time. For example, they bet high for 50% of trials with a 6 card, whereas participant 8 only bet high on the 6 card once (Figure 3B). The other participants, shown in gray, also varied their betting strategies (Figure 3B), which resulted in varied cumulative earnings (Figure 3A). Figure 3C shows that participant 8 had shorter reaction times than participant 4 for all cards except for the 2 card. In fact, participant 4 had longer reaction times than any other participant except for the 2 card.

### 3.2. A Trend between Theta and Gamma Activity in Large-Scale Brain Networks and Gambling Performance

We found that participants had various levels of activity in each large-scale brain network by examining the average band power during the decision-making window (between “Show Player Card” to “Start Movement”) in the different networks as a function of cumulative earnings. Pearson’s correlations were computed to capture any potential relationships between average earnings ($) and average band power across participants. Overall, we found general trends in each network where band activity decreased or increased with performance (Figure 4). There was a positive trend between theta power in DAN as well as gamma power in DMN and FN. Conversely, there was a negative trend between theta power in DMN and DN as well as theta and DAN. However, out of the six combinations, only the gamma activity in DMN was significantly correlated to performance (Pearson’s correlation, *p* = 0.05, r = 0.63). After correcting for multiple comparisons across two frequency bands and three networks using Bonferroni correction, this relationship was no longer significant. Although none were statistically significant, this is preliminary evidence that trends exist between band activity in these large-scale brain networks and performance. The small sample size makes correlations sensitive to outliers, as a sample size design using ɑ = 0.05, β = 0.05, and r = 0.63 reveals a sample size of 27 is needed [35]. Given more participants, we are confident that these trends would be significant.

### 3.3. Connectivity Strengths within and between Large-Scale Brain Networks Correlate to Performance

To explore the differences between connectivity and performance, we divided participants into performance groups as follows: High earners were those whose average earnings across trials were more than the median ($2.81), and low earners were those whose average earnings across trials were less than the median. First, we found a statistical difference between the average reaction times and performance (Mann-Whitney U test with Bonferroni correction, *p* ≤ 1 × 10^−2^). As demonstrated in Figure 5, lower earners took more time to react during the decision-making window than higher earners for all card types. Furthermore, all participants reacted slower to the 2, 4, and 6 cards relative to how quickly they reacted to the 8 and 10 cards. Taking more time to react would give the participants more time to process their choices to make a decision. This extra time should be reflected by additional brain processes or connectivity between performance groups. Therefore, we hypothesized that the performance groups would engage in different connectivity within and between networks.

We then analyzed the theta and gamma band functional connectivity within and between networks to investigate if performance and behavior had neural correlations with connectivity. Connectivity strength, as defined in the Methods, was computed as the square root of the sum of the squares of the individual covariances computed pairwise for all regions within the network (for within-network connectivity) or between each pair of networks (for between-network connectivity). In the theta band, the connectivity strength between high and low earners of DMN–DMN, DMN–DAN, DMN–FN, and DAN–DAN were significantly different (Figure 6A, Bonferroni-corrected Mann–Whitney U test, *p* ≤ 1 × 10^−3^, power: DMN–DMN, 0.997; DMN–DAN, 0.910; DMN–FN, 0.535; DAN–DAN, 1). Specifically, high earners had higher connectivity strength between these networks than low earners in theta. This relationship was mostly consistent for the gamma band as well; high earners had significantly higher connectivity strength for DMN–DMN, DMN–DAN, DMN–FN, DAN–DAN, and DAN–FN (Figure 6B, Mann–Whitney U test with Bonferroni correction, *p* ≤ 1 × 10^−3^, power: DMN–DMN, 0.989; DMN–DAN, 0.969; DMN–FN, 0.9999; DAN–DAN, 1; DAN–FN, 0.969). In contrast, low earners had significantly higher connectivity strength for FN–FN (Mann–Whitney U test with Bonferroni correction, *p* ≤ 1 × 10^−3^: FN-FN 0.643).

To visualize the connectivity between low and high earners, we summarized these results as a graphical model seen in Figure 7, where the nodes represent the three networks (DMN, DAN, FN) while the thickness and color of the edges represent the connectivity strength. This figure highlights how high earners exhibit higher connectivity strength both between and within most networks for theta and gamma (excluding FN). Additionally, we calculated the sign of the connectivity to be positive across both theta and gamma for all pairs of networks across all participants, shown in Appendix A. This reveals that the power between networks moves in the same direction by either increasing or decreasing together during decision-making.

## 4. Discussion

Our initial research goals were as follows: (i) to describe the relationship between betting behavior and performance across participants, (ii) to identify frequency bands in which networks relate to gambling performance, and (iii) to describe the relationship between connectivity and gambling performance. We hypothesized that participants would perform better in an economic decision-making task if they could selectively pay attention to the relevant task stimuli and have better cognitive reasoning capabilities. Three relevant large-scale brain networks that modulate their activity with selective attention and cognitive reasoning are the DMN, DAN, and FN [4,5,6,7,12,18,36,37,38]. Additionally, we found that the theta and gamma frequency bands were most correlated with subject performance. These networks were hypothesized to encode information about the task and behavior through theta and gamma band powers. By examining intracranial electrophysiological data from these three large-scale brain networks during an economic decision-making task, we found preliminary correlations between the theta and gamma activity in these bands as well as significant functional connectivity differences within and between the networks based on performance.

Firstly, we found that betting behavior and gambling performance varied across participants based on their gambling habits. Indeed, participants’ betting strategies were reflected by their behavior throughout the session. The optimal strategy that would maximize their expected reward while minimizing variance (risk) for our gambling task would be to bet $5 (low) on the 2, 4, and 6 cards, and bet $20 (high) on 8 and 10 cards. Choosing to bet low on the 6 card is optimal because the participant looses the amount they bet if they lose. We found that participant 8 implemented the optimal strategy the entire time, which rewarded them with the highest cumulative and average earnings. They also reacted the fastest when choosing their bet. Conversely, participant 4, who had the lowest cumulative and average earnings, did not use an optimal strategy as evidenced by their varying bets on all cards but especially the 6 card. Similar conclusions have been made in our previous work, which concluded that varying levels of internal bias affected future betting by overruling cognitive reasoning or optimal betting [21]. Indeed, the performances of participants 8 and 4 reflect the strategies used by the other high and low performers, respectively. Here, we extended these differences between strategies used by the high and low performers by examining the large-scale brain network activity and connectivity through our preliminary analysis.

Performance variability among the participants suggests varying levels of cognitive reasoning [9], which was reflected in the activity and connectivity in the DMN, DAN, and FN, specifically in the theta and gamma frequency bands. These networks have all been implicated in various aspects of cognition. Moreover, successful cognition also relies on the transfer of information between these networks as well. For our task specifically, many cognitive tasks needed to be implemented by participants. These include but are not limited to, locating and remembering their card value (selective visual attention and working memory), understanding the relationship between their card and the likelihood of the remaining cards in the deck to occur (working memory and quantitative reasoning), and then picking a bet based on their conditions (quantitative reasoning and motor control).

The DMN is associated with cognitive task complexity and internal planning [39], making it a point of interest for us based on our gambling task. The DMN also relates to memory recall, in which prior experience is used to make decisions. Regions in the DMN deactivate when tasks are too complex [13,40] but activate when the task is just complex enough to perform [36] or routine enough to be performed automatically [41], particularly in broadband gamma [13]. In our task, this would be reflected by participants with greater gamma activity in DMN earning more, in that participants with a better understanding of the cards may reflexively select the card based on the optimal strategy that yields the best outcome. Conversely, low-performing participants had deactivated power in DMN, suggesting that the task was too complex for them because they did not use an optimal strategy.

Our other main finding was that the relationship between the connectivity within and between the networks, DMN, DAN, and FN, reflects the participants’ performance in the theta and gamma bands. Namely, participants who performed the gambling task well had higher connectivity with most networks in both gamma and theta. There are many examples of connectivity and performance. To begin, higher connectivity between the DAN and DMN has been associated with better performance, in terms of reaction time and accuracy, during a moving dot motion task [42]. Indeed, previous work in our lab has also found extensive connectivity between regions in DAN and DMN related to task performance [21]. Other work specifically decision-making has found a mixture of connectivity, including within and between DMN and FN, associated with monetary value and choice [43], as well as within DMN associated with task performance [41], and between DMN and FN with accuracy of a recollection task [44]. These examples, along with our findings, highlight the coordinated effort needed from large-scale brain networks to form and execute decisions.

We also found that the variation in gamma connectivity with performance in the FN is significantly correlated to is significantly correlated to DAN and DMN activity. The FN’s relationship with both networks suggests that it may work as a hub by modulating connectivity during different parts of the task [15]. This could also relate to working memory and cognitive demand, where Harding et al. found that FN connectivity was associated with directing and engaging other regions [16]. Future experiments could further explore the connectivity between the FN and DMN in gamma, where participants could learn how to bet more logically by updating memory through the FN [45] as well as increased network activity when a reward was received [37] to create an association. Hence, participants with higher connectivity between the FN and DMN and increased activity tended to earn more in our study. We did find that low performers had higher connectivity within FN than high performers in gamma. This suggests that FN connectivity to other networks rather than its intrinsic connectivity may regulate decision-making to enhance performance. In particular, FN is associated with conflict processing and attention management [38]. This would be particularly important for low performers, who tended to make varied decisions not based on an optimal strategy. Furthermore, Araujo et al. found that people who they identified as “higher risk seeking for losses” had higher connectivity in FN [22].

Finally, we also found that the relationship within and between the DAN, DMN, and FN was positive. In other words, they would co-activate or co-deactivate together. The relationship between “task-positive” networks (e.g., DAN and FN) with “task-negative” networks (e.g., DMN) was originally believed to be an opposing one [46]. However, this has been misleading [47]. Indeed, the directionality of the relationship between these networks was circumstantial. For example, Piccoli and colleagues observed positive connectivity between the DMN and DAN during the encoding and retrieval phase of a working memory task [48]. In our case, we attribute the positive relationship to the design of our task that combines autobiographical planning, visuospatial planning, and working memory to uniquely engage the DAN, DMN, and FN.

As with any project, limitations were present. The sample size was small as access to patients implanted with intracranial electrodes was challenging. Placement of the intracranial electrodes across epilepsy patients differed as coverage was determined solely for clinical purposes. Since the electrodes were placed across different regions of the brain to localize the seizure onset zone, our coverage varied. To minimize this effect, we only chose regions and networks that had at least 3 participants to contribute to the population. Any electrodes found to be located in pathological areas were removed from our analysis. A larger population would have also helped to separate the differences between participant behavior. Changes to the decision-making task would also have helped us distinguish epochs of interest. One aspect that should be considered is varying the location of cursor fixation. This would improve the participant’s baseline activity by preventing participants from automatically placing their cursor in the center of the screen before the fixation period starts. Another consideration would be to compare the activity in networks during the decision-making period and win/loss state (i.e., the show card period) to observe which networks were involved during reward and whether performance was impacted if information was passed within or between networks between trials.

## 5. Conclusions

In conclusion, our preliminary analysis revealed the complex relationship between large-scale brain networks during an economic decision-making task and performance. Our study is the first to connect the DAN, DMN, and FN simultaneously during an economic decision-making task. Future research should explore the modulation interaction between frequency bands in terms of connectivity within and between networks. Because these results are preliminary due to the small sample size, further research is necessary to understand the exact nature of the relationship between large-scale brain network activity, connectivity, and participant performance. Future work that could validate a causal relationship between these networks would be one in which Transcranial Stimulation (TMS) was used to dissociate the connectivity between either of the networks during gambling to observe how participants’ response differs from their uninterrupted bets. Our results would be particularly applicable to individuals with dysfunctional network activity, such as patients with traumatic brain injury or major depression disorder. Both cases are associated with altered decision-making abilities that interfere with everyday life, which could be addressed directly through TMS therapy to enhance their quality of life.

## Figures and Tables

**Figure 1 brainsci-14-00773-f001:**
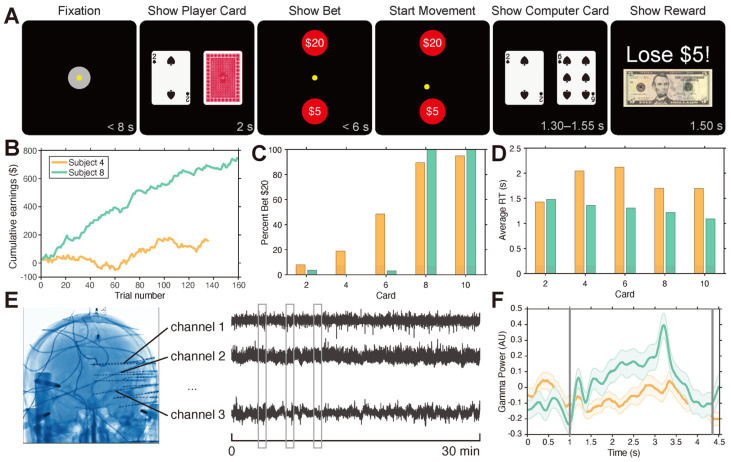
Betting behavior and performance varied across participants: (**A**) Timeline of visual stimuli shown to participants during an example trial for the decision-making task separated in epochs; (**B**) The cumulative earnings ($) across trials of the participants with the highest (participant 8, green) and lowest (participant 4, orange) average earnings; (**C**) Percentage of trials in which participants bet high ($20) for the participant with the highest (participant 8, green) and lowest (participant 4, orange) average earnings; (**D**). Average reaction time (RT) for the same participants; (**E**) Example electrode coverage (**left**) and recorded voltage traces (**right**) for *n* channels during a 30 min session; (**F**) Normalized gamma power in the cingulate cortex, averaged across all electrodes in the cingulate cortex and time-locked to “Show Player Card” (first vertical line) until the average onset of “Start Movement” (second vertical line) for the participant with the highest (participant 8, green) and lowest (participant 4, orange) average earnings. Solid lines represent the mean, and the shaded area represents 1 standard error.

**Figure 2 brainsci-14-00773-f002:**
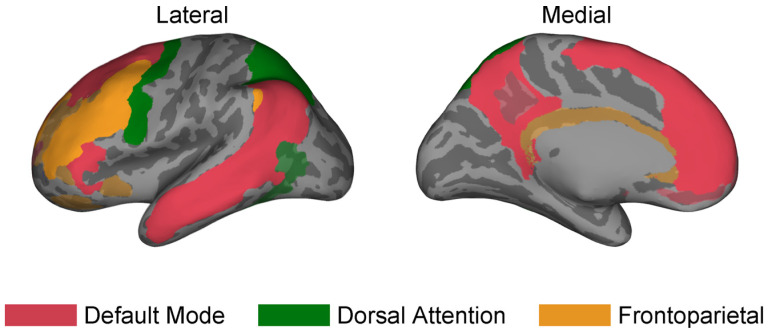
Summary of neural coverage. Regions are highlighted on an inflated template brain from FreeSurfer [34] based on the large-scale brain network they belong to, using the Yeo atlas [11]: Default Mode (DMN) in red, Dorsal Attention (DAN) in green, and Frontoparietal (FN) in orange. The dark gray and light gray represent the sulci and gyri, respectively. Only regions that had depth electrodes from at least one participant in this study are highlighted. The color-tinted grey areas represent other areas of the brain that each network belongs to but we did not have access to during our study.

**Figure 3 brainsci-14-00773-f003:**
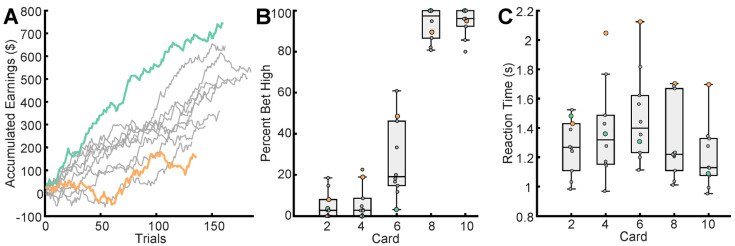
Summary of behavior during sessions: (**A**) Cumulative earnings ($) of participants across trials. The highest earner, participant 8 (green), and the lowest earner, participant 4 (orange), are highlighted in all panels; (**B**) Participants betting behavior based on each card. Percent of trials for which participants bet high ($20) across cards; (**C**) Participant’s reaction time (s) when betting based on each card. Note that the mean reaction time is longest for card 6 and shortest for card 10.

**Figure 4 brainsci-14-00773-f004:**
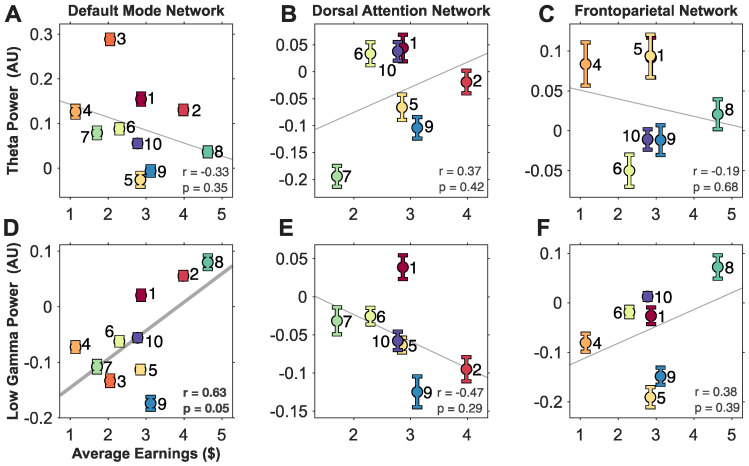
Average band power between large-scale brain networks and frequency bands. Scatter plots comparing the normalized power during decision-making (between “Show Player Card” to “Start Movement”) of large-scale brain networks (DMN (**A**,**D**), DAN (**B**,**E**), FN (**C**,**F**)) and in specified frequency bands (theta (**A**–**C**) and gamma (**D**–**F**)) across participants based on their average earnings ($). Each participant is represented as an error bar (mean ± 1 standard error) based on the distribution of power over average trial activity with a corresponding color and label. The solid gray line represents the best linear fit (least-squares). Pearson’s correlation was used between the power and average earnings, with corresponding r and *p*-value inset. A thicker linear fit line indicates a significant correlation (in (**D**)), all other fits are insignificant. After correcting for multiple comparisons using Bonferroni correction, none of the correlations remained significant.

**Figure 5 brainsci-14-00773-f005:**
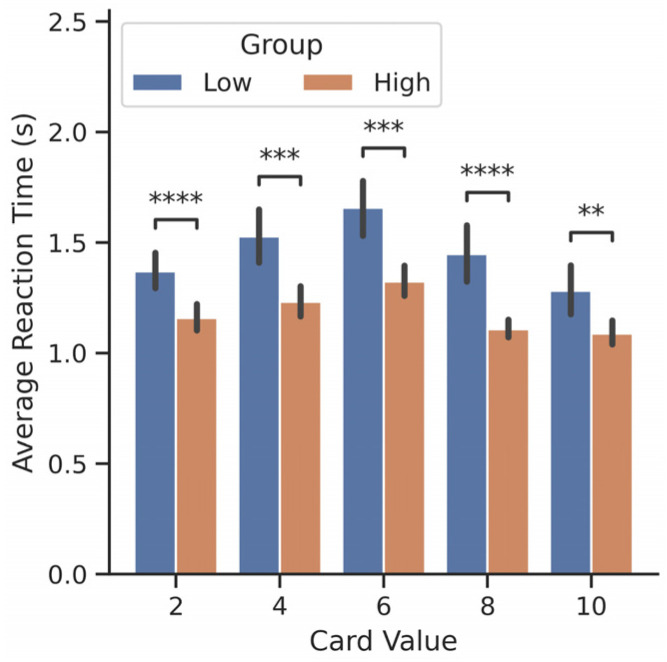
Reaction time comparison between earners. Average reaction time (seconds) of high (orange) and low (blue) earners based on their average earnings (USD, $) for each card value. Low earners took significantly longer to react than high earners. Statistics were performed with a Mann–Whitney U test, two-sided, with Bonferroni correction, with n = number of trials in each group, and *p*-value annotations are as follows: ns: not significant, *p* ≤ 1; **: 1 × 10^−3^ < *p* ≤ 1 × 10^−2^; ***: 1 × 10^−4^ < *p* ≤ 1 × 10^−3^; ****: *p* ≤ 1 × 10^−4^.

**Figure 6 brainsci-14-00773-f006:**
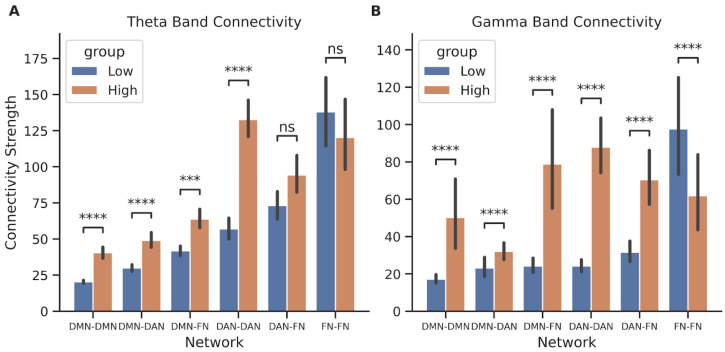
Theta and gamma network connectivity comparison between earners during decision-making. Bar plots comparing the average connectivity strength within and between networks across the high (orange) and low (blue) performing participants based on their average earnings ($) for (**A**) theta and (**B**) gamma. Statistics were performed with a Mann–Whitney U test, two-sided, with Bonferroni correction, with n = number of trials in each group, and *p*-value annotations are as follows: ns: not significant, *p* ≤ 1; ***: 1 × 10^−4^ < *p* ≤ 1 × 10^−3^; ****: *p* ≤ 1 × 10^−4^. See Appendix A for the complete power analysis.

**Figure 7 brainsci-14-00773-f007:**
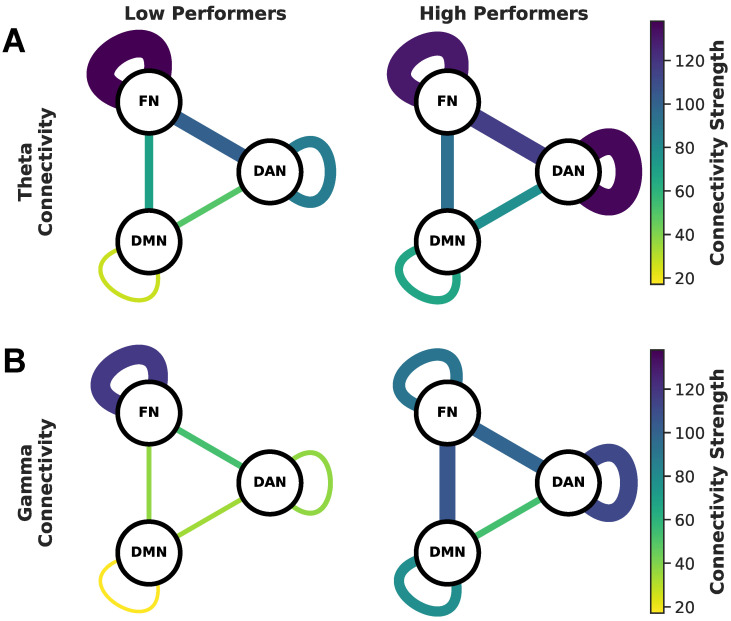
Summary of connectivity within and between networks. Connectivity strength within and between networks for low and high earners are summarized as network connectivity plots for (**A**) theta and (**B**) gamma. The line thickness and color of the lines connecting networks correspond to the strength of connectivity. Low earners are shown in the left column and high earners in the right column. Theta band is shown in the top row and gamma band is shown in the bottom row.

**Table 1 brainsci-14-00773-t001:** Demographics, clinical characteristics, and electrode coverage for each participant *.

Participant	Sex	Age	Handedness	EZ	DMN	DAN	FN
1	M	26	L	Hippo L, STG L	PCC (2), MTG (9), PC (2), SFG (1), STS (3)	IPS (1), SPL (2)	IPL (3), MFG (4)
2	F	41	R	Hippo R, EC R, TP R, PCC R, STG R	AG (4), PCC (3), F3o (2), ITS (2), MTG (6), PC (2), STS (4)	IPS (2), SPL (3)	
3	F	31	R	Hippo R	AG (4), PCC (1), F3o (6), F3t (4), ITS (1), MTG (8), PC (2), STS (3)		
4	F	53	R	INSa R	ACC (5), PCC (3), F3o (1), F3t (1), ITS (1), MTG (9), mPFC (2), STS (1)		IFS (1), MFG (3)
5	F	60	R	TP L	AG (7), PCC (3), MTG (11), PC (4)	IPS (3)	IPL (1)
6	F	36	R	PC R	AG (3), ACC (3), PCC (3), MTG (1), PC (7), SFS (1)	IPS (5), PRG (2), SPL (2)	IPL (5)
7	F	23	R	TP R, Hippo L/R	AG (5), PCC (4), ITS (1), MTG (14), PC (3), STS (1)	IPS (2)	
8	M	32	R	Hippo L	PCC (2), MTG (9)		IPL (2)
9	F	32	R	PCC L	AG (4), F3t (2), ITS (2), MTG (21), PC (8), STS (3)	IPS (1), SPL (2)	MFG (7)
10	M	28	R	Hippo L	ACC (3), PCC (3), F3o (5), F3t (5), MTG (5), PC (5), SFG (5)	IPS (1)	MFG (7)

* Table includes sex, age at time of recording (in years), epileptogenic zone (EZ), and location (number) of intracranial depth electrodes. Default mode network (DMN), dorsal attention network (DAN), frontoparietal network (FN). Angular gyrus, AG; cingulate cortex (anterior), ACC; cingulate cortex (posterior), PCC; entorhinal cortex, EC; hippocampus, Hippo; inferior frontal gyrus (pars orbitalis), F3o; inferior frontal gyrus (pars triangularis), F3t; inferior frontal sulcus, IFS; inferior parietal lobule, IPL; inferior temporal sulcus, ITS; intraparietal sulcus, IPS; middle frontal gyrus, MFG; middle temporal gyrus, MTG; precentral sulcus, PRG; precuneus, PC; prefrontal cortex (mesial), mPFC; superior frontal gyrus, SFG; superior frontal sulcus, SFS; superior parietal lobule, SPL; superior temporal gyrus, STG; superior temporal sulcus, STS; temporal pole, TP. Right, R; left, L.

**Table 2 brainsci-14-00773-t002:** Performance-related information for each participant *.

Participant	Total Trials	Won (%)	Lost (%)	Draw (%)	Cumulative Earnings ($)	Average Earnings ($)	Reaction Time (s)
1	185	40.00	42.70	17.30	530	2.86	1.01 ± 0.21
2	162	45.06	35.80	19.14	645	3.98	1.22 ± 0.38
3	144	36.81	39.58	23.61	295	2.05	1.63 ± 0.81
**4**	**136**	**37.50**	**45.59**	**16.91**	**155**	**1.14**	**1.85 ± 1.05**
5	172	37.79	48.84	13.37	490	2.85	1.10 ± 0.33
6	157	35.67	41.4	22.93	360	2.29	1.48 ± 0.63
7	132	37.88	37.8	24.24	225	1.70	1.19 ± 0.35
**8**	**160**	**42.50**	**34.38**	**23.13**	**740**	**4.63**	**1.28 ± 0.40**
9	154	40.26	36.36	23.38	480	3.12	1.37 ± 0.56
10	182	35.71	44.51	19.78	505	2.77	1.22 ± 0.43

* Total number of trials performed, percent of trials won, percent of trials lost, percent of trials drawn, cumulative earnings in USD ($), average earnings in USD ($), and reaction time (mean ± 1 standard deviation) in seconds. Participants with the highest (8) and lowest (4) cumulative earnings are bolded.

## Data Availability

The raw data supporting the conclusions of this article will be made available by the authors upon request.

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
