# Peer review of "An Exploratory Study of Large-Scale Brain Networks during Gambling Using SEEG"

_brainsci, 2024, doi:10.3390/brainsci14080773_

Round 1
Reviewer 1 Report
Comments and Suggestions for Authors
The authors researched the relationship between activity and connectivity of large-scale brain networks, that is, Default Mode Network (DMN), Dorsal Attention Network (DAN), and Frontoparietal Network (FN), in the context of performance during economic decision-making, i.e., a gambling task.
This is an exciting research report. In my opinion, the manuscript is well-handled. The research topic is actual and appropriate for the Brain Sciences journal. The experimental setup seems reproducible, and it is well-conducted. The discussion is well-written. Also, the limitations are clearly stated. The references are relevant.
Before the publication, I have the following comments/concerns for the authors to address
- L210; How did you suppress 60 Hz frequency? What filter or method did you use?
- L212; What window to suppress band leakage did you use (Hamming, Keiser, Blackman, Barlett-Han, ...)?
- L272; L280; Since you used a nonparametric statistical test (Mann-Whitney-Wilcoxon U test), in my opinion, a Spearman rank correlation test would be more appropriate to capture any potential relationships between variables.
Other than that, well done.
Author Response
The authors researched the relationship between activity and connectivity of large-scale brain networks, that is, Default Mode Network (DMN), Dorsal Attention Network (DAN), and Frontoparietal Network (FN), in the context of performance during economic decision-making, i.e., a gambling task.
This is an exciting research report. In my opinion, the manuscript is well-handled. The research topic is actual and appropriate for the Brain Sciences journal. The experimental setup seems reproducible, and it is well-conducted. The discussion is well-written. Also, the limitations are clearly stated. The references are relevant.
Authors: We thank the reviewer for their helpful comments and positive feedback. As detailed below, we have revised the manuscript to clarify the methodology. In addition, we provide a point-by-point response to each of the reviewer’s specific comments and indicate where we revised the manuscript.
Before the publication, I have the following comments/concerns for the authors to address
Comment 1: L210; How did you suppress 60 Hz frequency? What filter or method did you use?
Response 1: Thank you for your comment. Indeed, 60 Hz noise and its harmonics (120 Hz) were filtered out using a second-order Butterworth notch filter before bandpass filtering specific frequency bands. Our new manuscript includes the description of these filters:
- Line 239: “Additionally, 60 Hz electrical noise and its harmonic (120 Hz) were filtered out using a second-order Butterworth notch filter.”
Comment 2: L212; What window to suppress band leakage did you use (Hamming, Keiser, Blackman, Barlett-Han, ...)?
Response 2: We used a Hamming window to suppress band leakage, which can now be found in the manuscript:
- Line 246: “A Hamming window was applied to suppress band leakage.”
Comment 3: L272; L280; Since you used a nonparametric statistical test (Mann-Whitney-Wilcoxon U test), in my opinion, a Spearman rank correlation test would be more appropriate to capture any potential relationships between variables.
Response 3: Thank you for your suggestion. Upon implementing your idea, we found that the results yielded less significant results (See attachment). This may be due to the fact that ranking performance loses information on the actual differences between subject performances. Therefore, we decided to stick with our original analysis. As now stated in the title and in the discussion - this is an exploratory study and future work is to collect more subjects.

Reviewer 2 Report
Comments and Suggestions for Authors
“Exploration of Large-Scale Brain Networks during Gambling”( brainsci-3083451)
This manuscript aimed to explore the record neural activity from the default mode network (DMN), dorsal attention network (DAN), and frontoparietal network (FN) in humans while they performed a gambling task in the form of the card game “War” using SEEG. The results revealed that gamma band activity is directly related to a subject's ability to bet logically, deciding what betting amount will result in the most fiscal gain or least fiscal loss throughout a session of the card game war. We also found connectivity between the DAN and the relation to a subject's performance. Specifically, subjects with higher connectivity between and within these networks had higher earnings. These findings suggested that connectivity and activity between DMN, DAN and FN networks is essential during decision-making. Overall, this topic is interesting and important and the findings hold great theoretical and practical implications, which help to elucidate the neuro underpinnings of decision making. This manuscript is well-written and well-organized. However, some concerns appeared after reading the whole manuscript.
1. Although you mentioned that “In principle, our study is the first to connect the DAN, DMN, and FN simultaneously during an economic decision-making task.”, the DAN, DMN, and FN was summarized separately in the current introduction part. Thus, the previous literature review about the studies investigating two of them simultaneously was lacking and should be provided in the introduction part.
2. There are many other neuroscience methods (such as fmri and TMS) to investigate this topic in previous studies, then what are the advantages of sEEG need to be emphasized in the introduction part to help the readers get better understanding the reasons why you chose this specific method.
3. The following reference using sEEG is directly related to the current investigation and needs to be reviewed and discussed.
Shih, W. Y., Yu, H. Y., Lee, C. C., Chou, C. C., Chen, C., Glimcher, P. W., & Wu, S. W. (2023). Electrophysiological population dynamics reveal context dependencies during decision making in human frontal cortex. Nature communications, 14(1), 7821.
4. How did you determine the sample size? Did you calculate the sample size needed before formal study?
5. All figures should be provided with high-resolution version.
6. As the sample size is really small in the current investigation, thus, the preliminary nature of this investigation should be reflected in the title. And “sEEG”should also be included in the title.
Comments on the Quality of English LanguageMinor editing of English language required
Author Response
This manuscript aimed to explore the record neural activity from the default mode network (DMN), dorsal attention network (DAN), and frontoparietal network (FN) in humans while they performed a gambling task in the form of the card game “War” using SEEG. The results revealed that gamma band activity is directly related to a subject's ability to bet logically, deciding what betting amount will result in the most fiscal gain or least fiscal loss throughout a session of the card game war. We also found connectivity between the DAN and the relation to a subject's performance. Specifically, subjects with higher connectivity between and within these networks had higher earnings. These findings suggested that connectivity and activity between DMN, DAN and FN networks is essential during decision-making. Overall, this topic is interesting and important and the findings hold great theoretical and practical implications, which help to elucidate the neuro underpinnings of decision making. This manuscript is well-written and well-organized. However, some concerns appeared after reading the whole manuscript.
Authors: We thank the reviewer for their positive feedback and useful comments. As detailed below we have revised the introduction and discussion based on your feedback. In addition, we provide a point-by-point response to each of the reviewer’s specific comments and indicate where we revised the manuscript. We believe these edits enhance the clarity of our work.
Comment 1: Although you mentioned that “In principle, our study is the first to connect the DAN, DMN, and FN simultaneously during an economic decision-making task.”, the DAN, DMN, and FN was summarized separately in the current introduction part. Thus, the previous literature review about the studies investigating two of them simultaneously was lacking and should be provided in the introduction part.
Response 1: Thank you for your suggestion. We agree that it would be appropriate to discuss DAN, DMN, and FN as pairs in the introduction. Our new version of the manuscript now contains a paragraph elaborating on examples of relationships we found:
- Line 78: “Together, these networks orchestrate their activity through connectivity and intermediate networks culminating into behavior, particularly during decision-making. For example, all three networks have been found to modulate their activity individually and collectively through connectivity based on the dynamics of an internal state as dictated by task performance [22]. Studies using resting-state functional connectivity have identified increased connectivity between DMN and FN in individuals who are greater risk-seekers, highlighting the importance of their relationship during affective decision-making [42]. Interestingly, disruption in connectivity between these regions has been identified in people with traumatic brain injury, which is marked by deficient higher-order cognitive functions such as decision-making [46].”
Comment 2: There are many other neuroscience methods (such as fmri and TMS) to investigate this topic in previous studies, then what are the advantages of sEEG need to be emphasized in the introduction part to help the readers get better understanding the reasons why you chose this specific method.
Response 2: Thank you for your suggestion. Our new version of the manuscript now elaborates on the disadvantages of other techniques and the advantages of SEEG in the introduction:
- Line 91: “Most studies investigating the activity and connectivity of these networks have relied on non-invasive imaging techniques such as fMRI. However, the pitfall of fMRI lies in its lack of temporal and spatial resolution [47]. Modern recording techniques, such as stereoelectroencephalography (SEEG), though invasive, are superior in both categories by recording close to the neuronal source with high temporal resolution.”
Comment 3: The following reference using sEEG is directly related to the current investigation and needs to be reviewed and discussed.
Shih, W. Y., Yu, H. Y., Lee, C. C., Chou, C. C., Chen, C., Glimcher, P. W., & Wu, S. W. (2023). Electrophysiological population dynamics reveal context dependencies during decision making in human frontal cortex. Nature communications, 14(1), 7821.
Response 3: Thank you for this recommendation of very relevant literature. Though we did not capture OFC, their work and use of SEEG during decision-making warrants its inclusion in our paper. As such, we have integrated this paper into our introduction when discussing the advantages of using SEEG over fMRI:
- Line 96: “Of note is a recent study that used the enhanced spatial and temporal resolution of SEEG to uncover how subjective value during decision-making is represented by gamma activity in the orbitofrontal cortex [48].”
Comment 4: How did you determine the sample size? Did you calculate the sample size needed before formal study?
Response 4: Thank you for your inquiry. Using a sample size analysis, we would need a sample size of 27 to make significant claims. However, due to the nature of this type of data (human, clinical, depth electrodes), we do not have the luxury of obtaining those numbers. Despite this, we believe that this data can make important contributions. We note the limitation of our sample size numerous times in the paper and extensively in the discussion.
Comment 5: All figures should be provided with high-resolution version.
Response 5: We agree that the quality of some of our figures could be improved, most notably Figure 1. The updated version of the manuscript now contains these high-resolution figures.
Comment 6: As the sample size is really small in the current investigation, thus, the preliminary nature of this investigation should be reflected in the title. And “sEEG” should also be included in the title.
Response 6: Based on your suggestion, we have adapted the title to better reflect the scope of our investigation and emphasize our use of SEEG.
Line 2: “An Exploratory Study of Large-Scale Brain Networks during Gambling using SEEG”
Reviewer 3 Report
Comments and Suggestions for Authors
I thank the authors for giving me the opportunity to read this interesting work. Although it is a preliminary study and there are many similar works in the literature, it can serve as an interesting starting point.
Given the small number of subjects, I would "lighten" the title by calling it an "exploratory study." In the abstract, I would also mention the small sample size.
In the last part of the introduction, I would better clarify the objectives of the study, specifying the research questions rather than the task procedure, which I would explain more thoroughly in the procedure section.
In Table 1, I would include only the title and, under the notes section, provide disambiguations for the acronyms. I would try to find a way to explain each case in a more discursive and comprehensible manner.
I would ask to improve the graphic quality of the task images with better resolution.
For Table 2, apply the same approach as Table 1, with explanations in the notes.
I would address the discussions with the objectives.
For future perspectives, I would include the applicability and possible validation of the task.
Best Regards
Author Response
I thank the authors for giving me the opportunity to read this interesting work. Although it is a preliminary study and there are many similar works in the literature, it can serve as an interesting starting point.
Authors: We thank the reviewer for their positive feedback and useful comments. As detailed below we have revised the introduction and discussion based on your feedback. In addition, we provide a point-by-point response to each of the reviewer’s specific comments and indicate where we revised the manuscript.
Comment 1: Given the small number of subjects, I would "lighten" the title by calling it an "exploratory study." In the abstract, I would also mention the small sample size.
Response 1: Thank you for your suggestions. Though our sample size is substantial in similar literature pertaining to human recordings, we agree that we should be upfront with our sample size in the title and abstract. As a result, we’ve adapted our title to include “exploratory study” and added the number of subjects into the abstract.
- Line 2: “An Exploratory Study of Large-Scale Brain Networks during Gambling using SEEG”
- Line 16: “Here, we used SEEG (stereoelectroencephalography) to record neural activity from the default mode network (DMN), dorsal attention network (DAN), and frontoparietal network (FN) in ten humans while they performed a gambling task in the form of the card game “War”.”
Comment 2: In the last part of the introduction, I would better clarify the objectives of the study, specifying the research questions rather than the task procedure, which I would explain more thoroughly in the procedure section.
Response 2: Thank you for your constructive feedback. As such, we believe our edits to our manuscript better outline our research questions and objectives:
- Line 112: “Specifically, we were interested in investigating the following: (i) describe the relationship between betting behavior and performance, (ii) identify frequency bands in which networks relate to gambling performance, and (iii) describe the relationship between connectivity and gambling performance.”
Comment 3: In Table 1, I would include only the title and, under the notes section, provide disambiguations for the acronyms. I would try to find a way to explain each case in a more discursive and comprehensible manner.
Response 3: Thank you for your suggestion on how to improve our table. As such, we moved our abbreviations to the notes as opposed to the title. However, we believe that it is important that we are as transparent as possible about the regions and number of contacts that each subject contributed to each network. This table resembles how we conveyed this information in our previous publications.
Comment 4: I would ask to improve the graphic quality of the task images with better resolution.
Response 4: We agree that the quality of Figure 1 could be improved. As such, the updated version of the manuscript now contains figures with higher quality.
Comment 5: For Table 2, apply the same approach as Table 1, with explanations in the notes.
Response 5: Just as before, we moved the description to the footer of the table, leaving only the title at the top. We hope this aligns more with the formatting of Brain Science.
Comment 6: I would address the discussions with the objectives.
Response 6: Thank you for your suggestion. In line with your previous suggestion of outlining our objectives, we have made several changes to the discussion to align with these objectives. We believe that setting up our introduction with objectives enhanced our discussion. Below are a few examples of how we improved the discussion:
- Line 391: “(i) to describe the relationship between betting behavior and performance across subjects, (ii) to identify frequency bands in which networks relate to gambling performance, and (iii) to describe the relationship between connectivity and gambling performance.”
- Line 399: “Additionally, we found that theta and gamma frequency bands were the most relevant for conveying subject performance.”
- Line 406: “Firstly, we found that betting behavior and gambling performance varied across subjects based on their gambling habits.”
- Line 422: “Performance variability among participants suggests varying levels of cognitive reasoning [9], which was reflected in activity and connectivity in the DMN, DAN, and FN, specifically in the frequency bands theta and gamma.”
Comment 7: For future perspectives, I would include the applicability and possible validation of the task.
Response 7: Thank you for your comment. We decided to expand on the future direction of our discussion based on your feedback with the following material:
- Line 507: “Future work that could validate a causal relationship between these networks would be one in which TMS was used to dissociate the connectivity between either of the networks during gambling to observe how subjects’ response differs from their uninterrupted bets. Our results would be particularly applicable to individuals with dysfunctional network activity, such as patients with traumatic brain injury or major depression disorder. Both cases are associated with altered decision-making abilities that interfere with everyday life, which could be addressed directly through TMS therapy to enhance their quality of life.”
Round 2
Reviewer 3 Report
Comments and Suggestions for Authors
ok